Effect of the flavonoid baicalin on the proliferative capacity of bovine mammary cells and their ability to regulate oxidative stress

Perruchot Marie-Hélène 1
Gondret Florence 1
Robert Fabrice 2
Dupuis Emilien 2
Quesnel Hélène 1
Dessauge Frédéric 1 frederic.dessauge@inra.fr
1 PEGASE, INRA, AGROCAMPUS OUEST , Rennes , France
2 CCPA Group , Janzé , France
Alvarez-Rodriguez Javier
Electronic publication date: 2019 Mar 5
Publication date: 2019
Volume: 7
Electronic Location ID: e6565
Received 2018 Nov 9; Accepted 2019 Feb 1
Copyright: © 2019 Perruchot et al.
Copyright year: 2019
Copyright holder: Perruchot et al.
License: This is an open access article distributed under the terms of the Creative Commons Attribution License, which permits unrestricted use, distribution, reproduction and adaptation in any medium and for any purpose provided that it is properly attributed. For attribution, the original author(s), title, publication source (PeerJ) and either DOI or URL of the article must be cited.
License URL: https://creativecommons.org/licenses/by/4.0/

Keywords: Mammary epithelial cell, Baicalin, Oxidative stress, Cell culture, Dairy cow

Funding: Animal Physiology & Livestock System Department of the French National Institute for Agricultural Research (INRA) and CCPA group This work was supported by the Animal Physiology & Livestock System Department of the French National Institute for Agricultural Research (INRA) and the CCPA group. The funders had no role in study design, data collection and analysis, decision to publish or preparation of the manuscript.

==============================
Background

High-yielding dairy cows are prone to oxidative stress due to the high metabolic needs of homeostasis and milk production. Oxidative stress and inflammation are tightly linked; therefore, anti-inflammatory and/or natural antioxidant compounds may help improve mammary cell health. Baicalin, one of the major flavonoids in Scutellaria baicalensis, has natural antioxidant and anti-inflammatory properties in various cell types, but its effects on bovine mammary epithelial cells (BMECs) have not been investigated.

Methods

Explants from bovine mammary glands were collected by biopsy at the peak of lactation (approximately 60 days after the start of lactation) (n = three animals) to isolate BMECs corresponding to mature secretory cells. Cell viability, apoptosis, proliferative capacity and reactive oxygen species (ROS) production by BMECs were measured after increasing doses of baicalin were added to the culture media in the absence or presence of H2O2, which was used as an in vitro model of oxidative stress.

Results

Low doses of baicalin (1–10 µg/mL) had no or only slightly positive effects on the proliferation and viability of BMECs, whereas higher doses (100 or 200 µg/mL) markedly decreased BMEC proliferation. Baicalin decreased apoptosis rate at low concentrations (10 µg/mL) but increased apoptosis at higher doses. ROS production was decreased in BMECs treated with increasing doses of baicalin compared with untreated cells, and this decreased production was associated with increased intracellular concentrations of catalase and NRF-2. Irrespective of the dose, baicalin pretreatment attenuated H2O2-induced ROS production.

Discussion

These results indicate that baicalin exerts protective antioxidant effects on bovine mammary cells. This finding suggests that baicalin could be used to prevent oxidative metabolic disorders in dairy cows.

Introduction

The early lactation period in dairy cows is marked by severe metabolic stress due to high energy demand of milk production and due to concomitant limited feed intake, thus resulting in the mobilization of body reserves and a high risk of metabolic (e.g., milk fever and ketosis) and infectious diseases such as mastitis (Sordillo & Aitken, 2009). High demand for energy and nutrients requires large amounts of molecular oxygen for aerobic metabolism, which is accompanied by increased rates of reactive oxygen species (ROS) production (Aitken et al., 2009). This excessive production of ROS during the periparturient period can overwhelm the systemic and organ-specific antioxidant defenses, resulting in development of oxidative stress (Bernabucci et al., 2005; Castillo et al., 2005; Sordillo et al., 2007) and various alterations in metabolic and cell survival mechanisms within the mammary gland (Piantoni et al., 2010). Therefore, oxidative stress has been implicated in many pathophysiological conditions of dairy cows (Allison & Laven, 2000; Harrison, Hancock & Conrad, 1984; Miller, Brzezinska-Slebodzinska & Madsen, 1993). The proper balance between oxidants and antioxidants is likely essential for maintaining dairy cow health and performance.

Various macro- and micronutrients are directly involved in regulating cellular function and health through their anti-inflammatory and/or antioxidant properties (Kim et al., 2004; Mueller, Hobiger & Jungbauer, 2010). Many bioactive compounds, such as flavonoids, that have anti-inflammatory or antioxidant properties have been found in edible medicinal plants. Among these plants, Scutellaria baicalensis (S. baicalensis) is one of the major medicinal herbs used for the treatment of various inflammatory diseases, hepatitis, tumors and diarrhea in East Asian countries. The flavones baicalin, wogonoside and their respective aglycones, baicalein and wogonin, are the main bioactive compounds in Scutellaria roots. In particular, baicalin is considered the most abundant glycoside that contributes to the quality of S. baicalensis’ bioactivity, and the anti-apoptotic, antioxidant, antitumor, anti-inflammatory and immune modulatory activities of baicalin have been observed in different cell types and animal models (Chou et al., 2003; Hsieh et al., 2007; Kim et al., 2004; Liu et al., 2008; Mueller, Hobiger & Jungbauer, 2010; Shapiro et al., 2009; Xu et al., 2010). In the mammary gland, particularly in mammary epithelial cells, the effects of baicalin have not been well described (Guo et al., 2014). Recently, baicalin has been suggested as a potentially useful therapeutic agent for the treatment of bacterial infections in a mouse model of mastitis. Cell culture systems that utilize extracellular sources of H2O2 are useful for studying the toxicity and cellular adaptive responses to oxidative stress (Gille & Joenje, 1992). Hence, the purposes of this study were (i) to investigate the effects of baicalin on cell viability, proliferation and oxidative stress in bovine mammary epithelial cells (BMECs); and (ii) to determine whether baicalin has a protective effect on BMECs by restoring the cell redox state after H2O2-induced oxidative stress. This is a first step in understanding the possible beneficial effects of S. baicalensis extract on increased milk production, which was recently observed during the first 60 days postpartum in dairy cows (Robert, Leboeuf & Dupuis, 2014).

Materials and Methods

All the animal procedures were discussed and approved by the CNREEA No. 07 (Local Ethics Committee in Animal Experiment of Rennes) in compliance with French regulations (Decree No. 2013-118, February 07, 2013).

Mammary tissue sampling

Lactating Holstein multiparous (third lactation) cows (n = 3) raised at the INRA experimental barn (UMR PEGASE, Le Rheu, France) were used for mammary tissue collection. The cows were milked twice daily before mammary tissue collection at the peak of lactation (after 60 DMI). Biopsies from the left and the right halves of the udder (one sample per quarter) were taken approximately halfway between the base of the teat and the dorsal body wall in a region containing a large amount of secretory tissue, according to a method adapted from Farr et al. (1996). Mammary biopsies were sliced into small explants (five mm3) for tissue digestion and cell isolation.

BMEC preparation and primary culture

Mammary explants were digested and dissociated as described by Perruchot et al. (2016). To ensure that the isolated BMECs corresponded to mature secretory cells, the cells were sorted by flow cytometry using fluorescein isothiocyanate (FITC) anti-rat IgG1 CD49f (α6 integrin) (ref. 13-097-246; Miltenyi Biotec, Bergisch Gladbach, Germany). Isotype controls were used for each antibody to eliminate nonspecific background fluorescence. Flow cytometric analysis was performed on a data set of 30,000 events (single cells) using a MACSQuant® Analyzer10 (Miltenyi Biotec, Bergisch Gladbach, Germany), and the data were analyzed using MACSQuantify analysis software (Miltenyi Biotec, Bergisch Gladbach, Germany). The results are expressed as percentages (dot plot analysis) (Fig. 1A).

Figure 1 Effects of baicalin on cell proliferation in bovine MEC.

A prerequisite to this study was to validate our experimental model, notably by ensuring that isolated MEC corresponded to mature secretory cells. Hence, cells were dispensed at 500,000 and incubated in darkness at 4 °C for 30 min with Fluorescein isothiocyanate (FITC) anti-rat IgG1 CD49f (α6 integrin). CD49f positive cells are gated in red (A). Next, bovine MEC were incubated with different concentrations of baicalin (0, 1, 10, 100, 200 µg/mL) for 24 h in triplicate. (B) Cell proliferation was measured during 16 h using BrDU cell incorporation. MEC were plated in 96 well-plates at a density of 5,000/well. Increased Mean Fluorescence Intensity (MFI) values denoted higher proliferation rates in MFI (×1,000). Data shown as mean ± S.E. *p < 0.05 versus control (B) Phase contrast images of MEC in culture incubated with zero µg/mL of baicalin (C) and 100 µg/mL of baicalin (D). Black bar stands for 50 µm.

The BMECs were cultured in 96-well plates at a density of 5,000 cells/100 μL/well in MGE-P epithelial cells growth medium containing 10% FBS, 1× penicillin/streptomycin, 0.25× insulin-transferrin-selenium (ITS) without lactogenic hormones according to methods described in Perruchot et al. (2016). All cell culture products were purchased from Sigma-Aldrich (Saint-Quentin Fallavier, France). All BMEC cultures were grown at 37 °C under 95% air and 5% CO2.

Cell viability, cell death and proliferation assays

Cell viability was determined using 3-(4,5-dimethylthiazol-2-yl)-2,5-diphenyltetrazolium bromide (MTT) colorimetric assay (Sigma-Aldrich, St. Louis, MO, USA) to assess the metabolic activity of cells. Briefly, confluent monolayers of BMECs were treated with increasing concentrations of baicalin (0, 1, 10, 100 and 200 µg/mL) for 24 h. The media were removed at the end of the treatment period, and BMEC monolayers were then exposed to 200 μL of MTT solution (0.5 mg/mL in PBS) for 2 h. After washing the cells with PBS, the formazan crystals were solubilized with DMSO (200 μL per well; Sigma-Aldrich Chimie) before the plates were measured at 540 nm using a Multiskan Spectrum microplate reader (Thermo Fisher Scientific, Waltham, MA, USA) equipped with a spectrophotometer. The results are presented as the mean ± standard error of the mean of triplicate assays from three independent experiments.

Cell death was assessed using annexin-V/PI (propidium iodide)-double labeling to visualize apoptotic and necrotic cells. An Alexa Fluor 488 Annexin-V/Dead Cell Apoptosis Kit (Thermo Fisher Scientific, Waltham, MA, USA) was used according to the manufacturer’s recommendations. Briefly, BMECs cultured with or without increasing doses of baicalin (0–200 µg/mL) during 24 h were incubated for 15 min with annexin-V-FITC (5 μg/mL) and PI (1 μg/mL) and then diluted in 400 μL of annexin-binding buffer. Cells were then analyzed by flow cytometry. The following staining controls were used: unstained cells, cells labeled with annexin-V-FITC alone (without PI) and cells labeled with PI alone (without annexin-V-FITC). For each sample, 2 × 104 events were analyzed.

To assess proliferative capacity, BMECs were cultured for 24 h with increasing doses of baicalin (from 0 to 200 µg/mL), then labeled using 10 μM BrdU per well and re-incubated overnight following the manufacturer’s instructions. The amount of labeling was quantified by measuring the absorbance at 370 nm using a scanning multiwell spectrophotometer (Mithras LB 940, Berthold Technologies, Oak Ridge, TN, USA) with a reference wavelength of 492 nm. Increased optical density (OD) values indicate a higher proliferation rate.

Oxidative stress assays

ROS production in BMECs cultured with different doses of baicalin (1–100 µg/mL) was determined by fluorimetric assay using a five μM carboxy derivative of fluorescein, carboxy-H (DCFDA), added to the medium. After 90 min or 24 h of incubation, respectively, the hydroxyl radicals produced by the cells were estimated by quantifying dichlorofluorescein (DCF) using a multidetection microplate reader (Mithras, LB 940) with excitation and emission wavelengths at 485 and 535 nm, respectively. Samples incubated with 50 U of PEG-SOD were used as negative controls to assess the specificity of the assay.

Hydrogen peroxide was also applied to primary BMECs as an in vitro oxidative stress model (Gille & Joenje, 1992). The cytotoxicity of H2O2 to BMECs was first established by incubating BMECs with increasing concentrations of H2O2 (0–1,000 µM) diluted in cell culture media. Cytotoxicity was estimated after 24 h of treatment as described above. Because a large decrease in cell viability was observed at the highest doses of H2O2, the following tests were performed with H2O2 concentrations below 200 µM. ROS production was monitored in BMECs exposed to increasing doses of H2O2 (0, 25, 50, 100 and 200 µM) as described above. Finally, the potential of baicalin treatment to protect BMECs against H2O2-induced oxidative stress was estimated by pretreating BMECs with increasing concentrations of baicalin (24 h) and then exposing the cells to H2O2 (0, 25, 50 or 100 µM) in culture media for an additional 24 h. Cell viability and ROS production were examined at the end of this period as described above.

Protein extraction and western blot analysis

Total proteins were extracted with Radioimmunoprecipitation buffer (Fisher Scientific Illkirch, Illkirch-Graffenstaden, France) from BMECs cultured for 24 h with or without increasing concentrations of baicalin (0, 5, 10 and 100 µM/mL). Protein concentrations were determined using a BCA Protein Assay Kit (Fisher Scientific Illkirch, Illkirch-Graffenstaden, France) according to the manufacturer’s instructions. The proteins were separated by SDS-PAGE on 4–12% SDS-polyacrylamide gels (NuPage 4–12% Bis-Tris, NP0323BOX, Invitrogen Life Technology, Berlin, Germany), transferred to polyvinylidene difluoride membranes (GE Healthcare Bio-Sciences AB, Uppsala, Sweden) and incubated overnight with primary antibodies against catalase (SC-50508, Santa Cruz) or nuclear factor (erythroid-derived 2)-like 2 (NFE2L2/NFR-2; SC-722; Santa Cruz Biotechnology, Santa Cruz, CA, USA). Beta-actin (A5441; Sigma Aldricy, St. Louis, MO, USA) was used as a control. Horseradish peroxidase-conjugated secondary antibody was used at 1:2,500, and chemiluminescence was visualized using an ECL Kit and an ImageQuant LAS4000 Biomolecular Imager digital imaging system (GE Healthcare, Velizy-Villacoublay, France).

Statistical analysis

The data were first tested for normality. Experiments were repeated three times. Data were analyzed by one-way analysis of variance (ANOVA) using the following model: yij = µ + timei + εij (y = viability, proliferation, ROS production, western blot data; µ = mean; i = baicalin dose and ε = residuals). Tukey’s post hoc pairwise analysis was used. Differences were considered significant at p < 0.05. All statistical analyses were performed using RStudio (RStudio Team, 2018).

Results

Effects of baicalin exposure on the viability and apoptotic and proliferative capacities of BMECs in primary culture

To study the effects of baicalin on BMECs, we chose to use primary BMECs isolated from bovine mammary glands. We digested mammary secretory tissue from lactating dairy cows and isolated adherent BMECs. Prior to use, we verified that the BMECs expressed the classical epithelial cell marker CD49f. Indeed, it is obvious from Fig. 1A that the primary BMECs used for this study were epithelial since 91% expressed CD49f.

Next, we evaluated baicalin cytotoxicity in culture by increasing baicalin doses and evaluating the number of viable cells. Viability was significantly enhanced with 10 µg/mL baicalin but was similar under control conditions and medium supplemented with one µg/mL baicalin. Higher doses of baicalin (100 and 200 µg/mL) dramatically lowered cell viability (Table 1). Specifically, cell viability was two-fold lower after exposure to 200 µg/mL baicalin for 24 h than under control conditions (p < 0.001). Apoptosis was indirectly evaluated through the expression of annexin-V in cultured BMECs. The proportion of apoptotic cells was lower (5% vs. 8%; p < 0.05) with 10 µg/mL baicalin but was similar under control conditions and medium supplemented with one µg/mL baicalin. Higher doses of baicalin resulted in a dose-dependent increase in the proportion of apoptotic cells (Table 1), with 1.6-fold more apoptotic cells with 200 µg/mL baicalin compared with control conditions. The effects of increasing doses of baicalin were also characterized in terms of BMEC proliferation after 24 h of culture (Fig. 1B). Although the lowest concentration of baicalin (1 µg/mL) slightly increased BMEC proliferation (+10%, p < 0.05) when compared with the control, large doses of baicalin (100 µg/mL and 200 µg/mL) in medium decreased cell proliferation (p < 0.05), with more than 50% fewer proliferative cells at the highest dose of baicalin (200 µg/mL) compared with the control. Phenotypic differences between BMECs cultured in the absence of baicalin (control) or with a high dose of baicalin (100 µg/mL) are shown in Fig. 1C, with lower cell density and more disrupted BMEC colonies in the cells treated with 100 µg/mL baicalin. Taken together, these results show that low concentrations (1 and 10 µg/mL) of baicalin have no or slightly positive effects on BMEC viability, apoptosis and proliferation, whereas large doses (100 and 200 µg/mL) of baicalin clearly lowered proliferation, impaired viability and increased apoptosis of BMECs.

Table 1 Effects of baicalin on cell viability and apoptosis in BMEC.

	Baicalin concentration (µg/mL)	
	0	1	10	100	200	
Viability (OD)	13330 ± 998	13563 ± 428	14130 ± 1113*	7586 ± 377*	6286 ± 655*	
Apoptotic cells (%)	7.86 ± 0.5	7.27 ± 0.9	5.11 ± 0.3*	9.73 ± 0.3*	12.81 ± 0.5*	
Notes:

BMECs were incubated with different concentrations of baicalin (0, 1, 10, 100, 200 µg/mL) for 24 h in triplicate. Data shown as mean ± S.E.

* p < 0.05 versus control.

Baicalin lowered ROS production in confluent BMECs

Acute exposure (90 min) of baicalin to confluent BMECs, independent of dose, was associated with a large decrease in the amount of ROS released by BMECs into the culture media (−80%, p < 0.001, Fig. 2A). Chronic exposure (24 h) to baicalin was similarly associated with reduced ROS production by BMECs (Fig. 2B) when compared to BMECs cultured in a medium without baicalin. However, the inhibitory effect of baicalin on ROS production was lower at high concentrations (100 and 200 µg/mL) than at low doses (1–50 µg/mL).

Figure 2 Effects of baicalin on oxidative stress in bovine MEC after two treatment duration.

The effects of baicalin treatment were evaluated under different concentrations (0–200 µg/mL) during 90 min (A) and 24 h (B). The Reactive Oxygen Species (ROS) production was determined by a fluorimetric assay using five μM of 2′,7′-Dichlorofluorescin diacetate (DCFDA) added to the different media. After incubation, the hydroxyl radicals (ROS) produced by the cells were estimated by quantifying oxidation fluorescent product 2′,7′-Dichlorofluorescein (DCF). The oxidative stress is expressed as light intensity in percentage of the MFI control (0 µg/mL baicalin). *p < 0.05 versus control.

To investigate the mechanisms that might be involved in the reduction of ROS production by BMECs when exposed to baicalin, we analyzed the intracellular amounts of catalase, a powerful antioxidant enzyme and of Nrf-2, a transcription factor that controls the expression of antioxidant genes. A subset of baicalin doses (5, 10 and 100 µg/mL) was tested on BMECs cultured for 24 h and compared to the control. Treatment of BMECs with baicalin caused a marked increase in catalase and Nrf-2 protein (Figs. 3A and 3B). The greatest effect was observed at 10 µg/mL baicalin, with a 3-fold increase in catalase and Nrf-2 in treated cells compared with control cells. Taken together, low doses of baicalin may improve oxidative stress in BMECs by lowering ROS production and activating antioxidant intracellular defenses.

Figure 3 Effects of baicalin on Nrf-2 and catalase in bovine MEC.

MEC were cultured for 24 h in medium without (control) or with different concentrations of baicalin (5, 10 and 100 µg/mL). The cells were lysed in MPER buffer for total protein extraction, and 15 μg of protein was analyzed using electrophoresis and Western blotting (Catalase (A) and Nrf 2 (C) antibodies). Each band was quantified using a molecular imager and for each treatment, Catalase (B, 64 kDa) and Nrf 2 (D, 61 kDa) data were normalized using actin (40 kDa) data. Experiments have been performed four times. *p < 0.05 versus control.

Baicalin pretreatment limits H2O2-induced ROS production in BMECs

To study the possible benefits of baicalin pretreatment of BMECs during oxidative stress, we used an in vitro model of H2O2-induced oxidative stress. We first evaluated the effect of H2O2 on BMEC viability to determine the optimal working concentration. The data presented in Fig. 4A show that BMEC viability was significantly affected after treatment with 200 µM H2O2. Next, we investigated ROS production by BMECs when increasing doses of H2O2 were added to the cell medium. An exponential relationship between ROS production and increasing doses of H2O2 was observed (Fig. 4B), with high doses of H2O2 (200 µM) resulting in a 6-fold increase in ROS production compared to zero µM H2O2. Finally, increasing doses of H2O2 were added to BMECs pretreated for 24 h with increasing doses of baicalin. Although baicalin was not able to inhibit H2O2-induced ROS production in BMECs, ROS production was significantly lower in pretreated BMECs than in untreated cells (Fig. 4C). The largest reduction was observed after treatment with the highest doses of baicalin (100 and 200 µg/mL), independent of H2O2 dose. Finally, improved viability after the H2O2 was observed in cells pretreated with baicalin (Fig. 4D) compared to untreated cells, with the highest improvement observed at five µg/mL baicalin pretreatment. Altogether, pretreatment of BMECs with baicalin reduced ROS production under normal and oxidative stress conditions and improved cell viability when exposed to an acute stressor.

Figure 4 Effects of baicalin pretreatment following H2O2 stimulation on ROS production in bovine MEC.

BMECs were cultured for 24 h in medium without (control) or with different concentrations of H2O2 at increasing doses (0, 25, 50, 100, 200, 500, 1000 µM) in triplicate. Data shown as mean ± S.E. (A) BMECs were plated in 96 well plates at a density of 5,000/well. MTT (25 μl of five mg/mL) was added to check the cell viability assay (Optical density, OD). *p < 0.05 versus control (B) The ROS production was determined by a fluorimetric assay. Data are expressed in percentage of the control. *p < 0.05 versus control (C) and (D) BMECs were pretreated with different concentrations of baicalin (BC, 0–200 µg/mL). After 24 h, they were exposed to different concentrations of H2O2 (0, 25, 50, 100 µM) in triplicate (C) or to 100 µM in triplicate (D). The ROS production (C) was determined by a fluorimetric assay. Data are expressed in arbitrary unit (OD ×1000). *p < 0.05 versus control. Cell viability assay was also assessed (D). Results without Baicalin and with 100 µM H2O2 represent 100% viability. *p < 0.05 versus control.

Discussion

The effects of baicalin have been largely studied in vitro in transformed cell types, demonstrating various antitumor, hepatoprotective, anti-inflammatory and antibacterial properties (Chen et al., 2012; Lin et al., 2014b; Wang et al., 2008; Yin et al., 2011; Yu, Pei & Li, 2015; Zheng et al., 2012). In mammary gland, studies have been dedicated to investigating the role of baicalin as a possible agent for the treatment of breast cancers, showing either no impact on cell viability (Zhou et al., 2017) or a concentration-dependent (50–200 µM) decrease in cell viability (Zhou et al., 2009) without affecting programed cell death in human breast cancer MCF-7 and MDA-MB-231 cells (Zhou et al., 2009). However, the potential beneficial effects of baicalin on nontumorigenic mammary cells remain unclear. Anti-apoptotic properties of baicalin have been described in a Staphylococcus aureus-induced mouse model of mastitis (Guo et al., 2014). In the present study, one µg/mL baicalin slightly increased proliferation (+16%) and 10 mg/mL baicalin increased cell viability (+6%) and decreased apoptosis (−35%) in BMECs cultured for 24 h. Similarly, Zheng et al. (2014) suggested that the addition of 1–10 µg/mL baicalin in culture media may have anti-apoptotic effects by increasing anti-apoptotic Bcl-2 protein expression and decreasing caspase-3 protein expression in PC12 cells (Zheng et al., 2014). Lin et al. (2014a) also showed that baicalin pretreatment inhibited mitochondria-mediated apoptosis in vivo at a dose of 1–100 mg/kg baicalin. In contrast, in MCF-7 or RAW 264.7 cell lines, low concentrations of baicalin decreased cell survival (50 µmol/L or 22 µg/mL) (Lee et al., 2015; Wang et al., 2008). This finding contrasted with the results of our study, in which high concentrations of baicalin (100 or 200 µg/mL) had deleterious effects on BMEC viability. Indeed, 24 h of incubation with the highest dose of baicalin (200 µg/mL) induced massive cell death (+60%) and decreased BMEC proliferation (−50%) compared with untreated cells. Other studies have reported similar contradictory effects of baicalin on cell features depending on the dose. For instance, a dose-dependent dual effect of baicalin on angiogenesis has been observed in chick embryos (Zhu et al., 2016), with increased cell proliferation in developing blood vessels at a low dose (10 µg/mL) but increased cell death at higher doses (five mg/mL).

Because the development of oxidative stress in dairy cows during the transition period generally results in various alterations in metabolic and cell survival mechanisms in the mammary gland (Piantoni et al., 2010), we next focused on the potential effects of baicalin on BMEC ROS production under either normal or challenged conditions. We observed that, irrespective of the baicalin dose, short (90 min)- and long (24 h)-term exposure of BMECs to baicalin rapidly decreased ROS production under normal culture conditions. Under normal conditions, cells are protected by a wide range of antioxidant mechanisms, which include intracellular enzymes such as superoxide dismutase (SOD) and catalase, to remove ROS (Schogor et al., 2013). This process is mediated by activation of the nuclear factor-erythroid-2-related factor 2 (Nrf2) a master regulator of the ROS response; its activation regulates the expression of genes that encode cellular defense enzymes and antioxidant proteins that contain an antioxidant response element (Cardozo et al., 2013). Ma et al. (2018) demonstrated that in isolated BMECs exposed to super-physiological doses of H2O2 (600 μM) for 6 h, the NFE2L2-ARE (NRF2-antioxidant response element) signaling pathway is a vital regulator of oxidative damage and inflammation (Ma et al., 2018). In our study, we showed that intracellular concentrations in catalase and Nrf2 were increased after baicalin treatment. In murine neuroblastomas, baicalin improved SOD activity and promoted the translocation of Nrf2 to the nucleus (Kensler, Wakabayashi & Biswal, 2007). Moreover, in a rat model of Alzheimer’s disease, baicalin treatment increased the activity and gene expression levels of antioxidant enzymes (SOD, catalase and glutathione peroxidase), and this increase was also associated with Nrf2 activation (Ding et al., 2015). Hence, the current study suggests that baicalin could prevent oxidative stress by decreasing ROS production through the Nrf2 pathway. However, direct evidence should be further provided by using transactivation assays.

In the present study, H2O2 was used as an in vitro model of oxidative stress (Gülden et al., 2010). H2O2 is a particularly important contributor to pathological events that, compared to superoxide anions, can cause intracellular and extracellular damage depending on the availability of reactive substrates (Multhaup et al., 1997; Yin et al., 2011). In the present study on BMECs, increasing doses of H2O2 stimulated ROS production that increased in an exponential manner. In proliferating mammalian cells, the following patterns of H2O2 responses have been described (Babich et al., 1996; Davies, 1999; Wiese, Pacifici & Davies, 1995): very low doses (3–15 μM) stimulated cellular growth, higher doses (120–150 μM) induced a temporary growth arrest, intermediate concentrations (250–400 μM) caused a permanent growth arrest, and high concentrations (≥one mM) induced necrotic cell death. In cancer models such as human breast adenocarcinoma cells (MCF-7), H2O2 also promoted damage such as DNA fragmentation and cell death (Dasari et al., 2006). Hence, cell culture systems that utilize extracellular H2O2 are especially useful to study the toxicity and cellular responses to oxidative stress. Importantly, the present findings suggest that pretreatment by baicalin protected BMECs from the oxidative stress induced by H2O2. Indeed, baicalin pretreatment decreased H2O2-induced ROS production by 50% and increased cell viability by 10%. Similar to our results in BMECs, pretreatment of MAC-T cells with resveratrol, a natural polyphenolic compound found in many plant species, limited the decrease in cell viability and prevented intracellular ROS accumulation observed after H2O2 exposure (Jin et al., 2016). Altogether, the results suggest that the antioxidant properties of baicalin may help protect the mammary epithelium under normal or challenged conditions.

Conclusions

In this study, we demonstrated that baicalin has positive effects on BMECs in vitro by regulating cell proliferation, apoptosis, cell viability and the antioxidant response and that these effects were generally observed at low concentrations of baicalin (1–10 µg/mM). Recently, dietary supplementation of dairy cows with S. baicalensis extract resulted in increased milk production during the first 60 days postpartum (Robert, Leboeuf & Dupuis, 2014). In vivo plasma concentrations of baicalin, which was originally administered as a food additive, were approximately 10 µg/mL. Taken together, we suggest the use of baicalin as a natural approach to promote daily cow lactation and health to minimize the negative effects of oxidative stress on dairy cow mammary glands during the peripartum period.

Supplemental Information

Supplemental Information 1 Uncropped electrophoretic gels and blots.

Actine, Catalase, Nrf-2.

Click here for additional data file.

The authors are grateful to the staff of the UMR PEGASE (INRA, Agrocampus Ouest, Saint Gilles, France), especially to Laurence Finot and Frédérique Mayeur for the laboratory analysis. The authors are grateful to American Journal Expert (Durham, NC, USA) for language editing.

Additional Information and Declarations

Competing Interests

Author Contributions

Animal Ethics

Data Availability

Marie-Helene Perruchot is an academic employee of the French National Institute for Agricultural Research, Florence Gondret is an academic employee of the French National Institute for Agricultural Research, Fabrice Robert is an employee of CCPA group, Emilien Dupuis is an employee of CCPA group, Helene Quesnel is an academic employee of the French National Institute for Agricultural Research and Frederic Dessauge is an academic employee of the French National Institute for Agricultural Research.

Marie-Hélène Perruchot conceived and designed the experiments, performed the experiments, analyzed the data, contributed reagents/materials/analysis tools, prepared figures and/or tables, authored or reviewed drafts of the paper, approved the final draft.

Florence Gondret conceived and designed the experiments, authored or reviewed drafts of the paper, approved the final draft.

Fabrice Robert authored or reviewed drafts of the paper, approved the final draft.

Emilien Dupuis authored or reviewed drafts of the paper, approved the final draft.

Hélène Quesnel authored or reviewed drafts of the paper, approved the final draft.

Frédéric Dessauge conceived and designed the experiments, performed the experiments, analyzed the data, contributed reagents/materials/analysis tools, prepared figures and/or tables, authored or reviewed drafts of the paper, approved the final draft.

The following information was supplied relating to ethical approvals (i.e., approving body and any reference numbers):

All the animal procedures were discussed and approved by the CNREEA No. 07 (Local Ethics Committee in Animal Experiment of Rennes) in compliance with French regulations (Decree No. 2013-118, February 07, 2013).

The following information was supplied regarding data availability:

Portail data Inra is available at:

https://data.inra.fr/dataset.xhtml?persistentId=doi:10.15454/EEBDJI.

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
