# Peer review of "Effect of the flavonoid baicalin on the proliferative capacity of bovine mammary cells and their ability to regulate oxidative stress"

_PeerJ, doi:10.7717/peerj.6565_

## Round 0.1 · original submission · Major Revisions

Dear authors,

In addition to reviewers' comments, it would be advisable to include the word 'flavonoid' somewhere in the title in order to set the framework of baicalin compound.

We look forward to hearing from you soon.

Reviewer 1 ·

Basic reporting

Clear, unambiguous, professional English language used throughout.
Intro & background to show context. Literature well referenced & relevant.
I suggest you to specify that the metabolic diseases “milk fever, ketosis” at line 45 are examples. I recommend (e.g., milk fever and ketosis).
Citations from the line 54 (Miller et al., 1993, Harrison et al., 1984, Allison and Laven, 2000) does not meet the editing requirements of the journal, please consider: (Miller et al., 1993; Harrison et al., 1984; Allison and Laven, 2000). Also, Miller et al. 1993, Harrison et al., 1984, and Allison and Laven, 2000 are missing in references list.
The name of Scutellaria at line 64 must be written in italics.
The reference (Robert et al., 2014) at line 79 must be written (Robert, Leboeuf & Dupuis, 2014)
The reference (Yu et al, 2015) at line 270 must be written (Yu, Pei & Li, 2015)
The reference Lin et al. at the line 282 don’t have the year in brackets.
The reference Loor et al. (2018) at line 305 is missing in references list.
The reference (Kensler et al. (2007) at lines 310-311 is missing in references list.
The reference (Wiese et al., 1995) at line 323 must be written (Wiese, Pacifici & Davies, 1995)
The reference (Jin et al., 2016) at lines 334-335 is missing in references list.
The Structure of References section is not conforms to PeerJ standards.
Structure conforms to PeerJ standards, discipline norm, or improved for clarity.
Figures are relevant, high quality, well labelled & described.
Raw data supplied.

Experimental design

Original primary research within Scope of the journal.
Research question well defined, relevant & meaningful. It is stated how the research fills an identified knowledge gap.
Rigorous investigation performed to a high technical & ethical standard.
Methods described with sufficient detail & information to replicate.

Validity of the findings

Data is robust, statistically sound, & controlled.
Conclusions are well stated, linked to original research question & limited to supporting results.

Additional comments

Many references in the text are missing in the References section.
This scientific article can be accepted but after revision

·

Basic reporting

no comment

Experimental design

no comment

Validity of the findings

no comment

Additional comments

Hello! Please pay attention to the modifications!

Reviewer 3 ·

Basic reporting

Briefly, the paper is scientifically valid, well written and clear. References are OK. Professional English Language is used and it is fluent.
Suggestions are below.


Fig 1 Legend, please be more concise for panel A;
“The reaction product was quantified by measuring the absorbance at 370 nm with a reference wavelength of 492 nm.”, this sentence could be removed.
“Minus Fluorescence Intensity (MFI)”, I guess the authors wanted to write “mean” or “median” instead of minus. Did they?

Fig 3 Legend, Random numbers are present in the text starting from this Figure, e.g. “…6 hydroxyl radicals”, “…7 dichlorofluorescein (DCF”)”, etc
“The fluorescence was measured with a multi-detection microplate reader using an 8 excitation and emission light at 485 nm and 535 nm, respectively.”, this sentence could be removed

Fig 4 Title, please correct H2O2 to H2O2, please remove the “3” before “production.

Line 131 “XX” hours. Please add the right time or rephrase
283 “in vivo” it should be italic
302 “H2O2”,
317 “in vitro”, it should be italic
343 “in vivo”, it should be italic

Experimental design

Material are very detailed and methodology are accurate and appropriate. The experiments seem clear and reproducible.

Labelling used for statistical significance is unordinary and may confuse the reader. I recommend to display the differences between the values in a different way and use asterisks for significance level.

Validity of the findings

The manuscript provides novel insights into modulation of mammalian antioxidant defences by baicalin. The research question is confined while sufficiently supported by the results. The discussion is well stated and coincise.

In the abstract the authors wrote “Baicalin prevented apoptosis at low concentrations (10 μg/mL)…”, but I suggest the authors to use a different verb.

Additional comments

Fig 1 panel C seems a bit uncomplete or uninformative as it stands. The filed is evolving and interesting. Improved in vitro models will help your future research and following impact.

---

## Round 0.2 · accepted · Accept

We acknowledge you for addressing the suggested comments in a satisfactory way.

#